# Basic Affective Systems and Sex Differences in the Relationship between Anger and Fear

**DOI:** 10.3390/ijerph21101266

**Published:** 2024-09-24

**Authors:** Paola Manfredi

**Affiliations:** Department of Clinical and Experimental Sciences, University of Brescia, 25123 Brescia, Italy; paola.manfredi@unibs.it

**Keywords:** sex differences, basic affective systems, SEEKING, FEAR, ANGER, SADNESS, PLAY, CARE, gender inequality

## Abstract

Background: The possible interactions between anger and fear have not been widely explored in the psychological literature. Fear and anger are currently beginning to be studied by looking at their interrelationships, rather than seeing them as simply opposing emotions. Furthermore, there is a tendency to think that anger is more typical of men and fear of women. Our contribution proposes a particular perspective of affective neuroscience. The objectives of the study are as follows: (1) to assess possible differences in affective systems, and states and traits of anger in relation to biological sex; (2) to assess correlations between ANGER, FEAR, and SADNESS, as well as state and trait anger in both a female and male sample; (3) to assess possible differences in basic affective systems in relation to different levels of ANGER, FEAR, and SADNESS, state and trait anger, in female and male samples. Methods: A non-clinical sample of 339 females and 99 males completed the ANPS 3.1 to assess basic affective states and the STAXI-2 to assess anger states and traits. Results: No significant differences were found for ANGER and FEAR scores and for state and trait scores between the male and female samples. Clear correlations emerged (*p* < 0.01) between SADNESS and FEAR in both the female and male samples. Among the differences that emerged in the affective systems, we emphasise that in the female group, the highest scores on the SEEKING and PLAY scales are expressed by the group of women who have the lowest scores in FEAR; PLAY and CARE also vary in relation to different scores in SADNESS. Discussion: Given the importance of the SEEKING and PLAY variables, it is of paramount importance to monitor the environmental and relational situations to guarantee that women, too, are provided with the conditions of safety and protection that are prerequisites for their well-being and the positive expression of their resources.

## 1. Introduction

### 1.1. Sex Differences in Anger and Fear

There is a cultural prejudice, partly also scientifically endorsed, that connotes anger as more masculine and fear as more feminine.

This prejudice is built on the assumption that anger would be more frequent in males, women express less intense anger than men [1,2,3], and men show a unique sensitivity to threatening stimuli, that is, to stimuli linked to angry emotions [4,5].

Brody et al. (1995), evaluating the changes in the developmental period from childhood to adulthood, found that in the adult sample, women tended to report more anger than men [6], but even recent studies, with new methodologies, seem quite aligned with the most historical and cited research. In an eye-tracking study, Zhang et al. (2024) report that “men’s total fixation duration and total fixation point number in angry emotions were significantly higher and greater than women’s, indicating that men have a unique attention bias toward threatening stimuli” [7].

In general, various studies have shown that there are gender differences in the neural networks for emotion processing [8,9] and that men have greater right hemisphere lateralization and higher activation levels than women [10,11].

With regard to fear, several studies showed that females feel more fear than males do [6,10,12,13], and recognise facial expressions of fear faster and more accurately than men [2,14]; women also invest more attention and resources on the target of fearful emotion and can notice it earlier [7]. The most widespread interpretation is that women report more fear due to a vulnerability that would depend on physical characteristics, as well as gender differences in neural networks for emotion processing and cultural aspects, i.e., due to their inferior status of power and the pressure of socialisation, through which women learn to be ‘the weaker gender’. According to Brody (1985) [6], therefore, fear is adaptive for women, because, on an intrapersonal level, it can protect them from the consequences of male aggression by intensifying coping strategies and, on an interpersonal level, it can serve to minimise aggression or warn others, e.g., children, that they are not safe. Even twenty-five years later, the interpretation of fear is not substantially different. Klein (2000) [15] also emphasises the female advantages in identifying negative emotions for the survival of the self and the offspring and recognises the role of parental investments in modulating responses to environmental threats to increase the survival of the self and the offspring. To think that being afraid is the best solution for a woman, who is in an inferior position, may well be true, but it is this inferior position that needs to be changed to achieve gender equality. There are undeniable differences between women and men, but there are conditions of inferiority that are attributable to culture, not nature, and changing them is therefore desirable.

Another underlying assumption, which is in our view questionable, is that anger is the alternative to fear and that the first is always preferable. From a relational point of view, the violent expressions of anger are precisely sustained by the intolerable perception of one’s own vulnerability; anger, therefore, can be used in its transformative function, turning one who is actually a sheep into a lion, so to speak [16]; expressing anger does not therefore always mean being in a position of superiority, but rather sometimes being unable to tolerate not being so.

### 1.2. The Interactions between Anger and Fear

The possible interactions between anger and fear have not been widely explored in the psychological literature. In fact, the prevailing view is that anger and fear are alternative responses. Both fear and anger involve the processing of threat signals [17], but anger promotes action and attack [18,19,20], whereas fear induces flight or freeze responses. This seems to be true both in the micro-dimension of inter-individual relationships and in the macro-dimension of conflicts between nations [21].

From a brain-related perspective, there are different activations, detailed as follows: anger activates a large cortical network [22], whereas fear is more related to the amygdala and subcortical circuits [23,24]. Furthermore, anger and fear responses to stress have different biological profiles, with anger being associated with a greater associated increase in cortisol, whereas fear responses are associated with an increase in pro-inflammatory cytokines [25].

However, there are also divergent contributions from fields as diverse as philosophy [26], neurology and medicine [27,28,29], psychology and psychopathology [30,31,32]. Anger and fear have also been studied in cases of traumatic experiences, as follows: the development of fear-based psychopathological forms triggered by trauma exposure also leads to changes in anger, such as excessive trait anger, anger expression and anger control deficits [30,31]. Marshall et al. (2018) [32] found a positive association between trauma exposure, fear, anger, intimate partner aggression (IPA) and parent–child aggression (PCA), but also differences related to gender and parental role.

### 1.3. This Study

Our contribution fits into the theoretical framework of affective neuroscience and thus sees fear and anger as part of the basic affective systems. The term affective neuroscience was coined by Panksepp [33], who identified through electrical brain stimulation seven basic affective systems located in primitive subcortical regions of the brain (periaqueductal gray (PAG), located in the midbrain) that are anatomically, neurochemically, and functionally homologous in all mammals. They are ancestral tools for living—evolutionary memories encoded in the genome in approximate form (as primary brain processes), which are subsequently refined by basic learning mechanisms (secondary processes) and higher-order cognitions/thoughts (tertiary processes).

Anger and fear are therefore both useful emotions for our survival, even though they may cause us pain. We cannot say that one is better than the other; both are present in males and females, and both require an adaptive response for good health.

The basic affective systems are intrinsic values that inform how we proceed in the pursuit of survival, i.e., they are ‘built-in tools for survival’, signalling whether we are in ‘comfort zones’, i.e., conditions that support survival, or in ‘discomfort zones’, indicating the presence of conditions that may compromise survival [33,34,35,36]. Contrary to how it may seem, positivity or negativity is not linked to the affective system; that is, we cannot say that CARE is positive and FEAR negative, but the affect is negative when it expresses an unsatisfied need or a lack of homeostasis; on the contrary, the affect is positive when it signals a return to a homeostatic state, subjectively perceived as pleasure.

Under healthy conditions, all affective systems provide an adequate response; on the contrary, under the condition of distress or psychopathology, there may be a hyperactivation of one system or a hypoactivation of another.

From a health perspective, we can then consider it relevant not only to consider the differences, if any, between anger and fear, in women and men, but it also seems more meaningful to us to consider the state of well-being, that is, to consider what happens to the other basic affective systems. Indeed, we believe that the activation of other affective systems can not only modulate the expression of anger and fear, but also allow us to hypothesise the implications and (clinical) consequences of certain activations.

Another element of interest in this approach is the identification of the PANIC/SADNESS system, which can give rise to emotions and behaviours that which may be superficially coded as fear, but which it is important to differentiate. FEAR requires, as an appropriate response, us to distance ourselves from the source of the fear, whereas PANIC/SADNESS has to do with the fear of abandonment, and this fear is therefore responded to by approaching or by seeking/waiting for the caregiver.

The box (Table 1) shows the salient features of the different affective systems [36].

The use of capital letters is to identify neural circuits, as opposed to the ordinary use of these same words.

The aims of the study are as follows:(1)To assess possible differences in affective systems, state, and trait anger in relation to biological sex.(2)To assess correlations between ANGER, FEAR and PANIC, as well as state and trait anger, in both a female and a male sample.(3)To assess possible differences in basic affective systems in relation to different levels of state and trait anger scores, ANGER, FEAR and PANIC, in female and male samples, in a non-clinical setting.

For the purposes of the research, also considering the theoretical framework of affective neuroscience, we have chosen to use the male/female division, but we are aware that this division is a dichotomous construct, based on a binary economy of the mental simplification of the objects of the world. We specify that we asked the subjects in the sample to indicate sex and not gender. “Gender interacts with sex but is different from it, which refers to the different biological and physiological characteristics of females, males and intersex persons, such as chromosomes, hormones and reproductive organs. Gender and sex are related but different from gender identity” [37].

## 2. Materials and Methods

The study was approved by the Provincial Ethics Committee of Brescia, n°3676, and by the Director General of the Municipality of Brescia.

The questions were proposed to the employees of the municipality of Brescia, anonymously, through granting access to a link for the completion of the questionnaire. The sending of the link was preceded by a letter written by the researcher, introducing the project and providing an e-mail and telephone number for further clarification, as well as a letter signed by the Director General informing employees of the research and inviting them to participate. The access link to the questionnaires was sent by the municipal administration, so as not to pass the institutional e-mails on to external staff. The completion of the questionnaires was only visible to the researcher, without them being able to know the e-mail and identity of the person completing them. Participation in the study was voluntary and employees could withdraw from the study without penalty or prejudice.

Inclusion criteria: between 20 and 68 years of age, no outstanding criminal convictions for violent crimes, and in a sufficiently good psycho-physical condition. The last two conditions were implied if the subjects were able to work. Age was also requested when filling in the questionnaires.

The subjects gave their informed consent in digital form.

### 2.1. Instruments

#### 2.1.1. ANPS 3.1

The Affective Neuroscience Personality Scale (ANPS) is a testing instrument that allows for the application of affective neuroscience principles to human clinical studies and examines individual differences in basic emotional systems. It should be pointed out that in this theoretical context, emotions are to be understood in a special sense. As is effectively underlined by Alcaro [38], the emotions are not so much feelings about what has happened in the body, but rather feelings about what is going to happen or might happen within a field that includes the organism and its environment. Emotions, in fact, have an anticipatory function; they orient behaviour along stemming from o-erect behaviour along particular adaptive directions that have been preserved in the course of natural evolution. Thus, affects are primarily directions of meaning; indeed, they are the primary directions of the meaning of consciousness.

The Affective Neuroscience Personality Scale, first published by Davis, Panksepp and Normansell in 2003 [39], and revised by Davis and Panksepp in 2011 [40], is a rating scale for basic affective states. The authors of the scale aimed to access personal feelings and behaviours rather than more cognitive social judgments [28,39], and consider the ANPS an instrument for indirectly assessing one’s emotional nature in the context of personality. Davis and Panksepp [40] claimed to interpret the ANPS scales as tertiary (thought-mediated) approximations of the influence of various primary emotional systems in people’s lives. ANPS 3.1 is the most recent version of the scale created by Panksepp and Davis [41]. This scale was translated into Italian by two researchers, with the permission of its author (Ken Davis, the only living author); the versions were compared to arrive at a final version, which was submitted to an English-speaking expert and compared with the English version. Responses on a 5-point Likert scale indicate the degree of agreement or disagreement with the statements proposed.

The ANPS assesses six primary emotions (excluding LUST) with 14 items each; there are also 12 additional items assessing spirituality, which are not included in this study. The scale also assesses (Social) Dominance and Social Anxiety. These are not the basic affective systems, so unlike the six primary emotions, which must be written in upper-case letters, they appear in lower-case letters. Examples of items are given as follows: “Almost any small problem or puzzle stimulates my interest” (SEEKING); “I often worry about the future” (FEAR); “I often feel a strong need to take care of others” (CARE); “When I am frustrated, I usually get angry” (ANGER); “I am known as someone who makes work fun” (PLAY); “I tend to think a lot about losing loved ones” (SADNESS). In our sample, the Cronbach’s alpha values are as follows: SEEKING 0.7067; FEAR 0.8719; CARE 0.7565; ANGER 0.8197; PLAY 0.8246; PANIC 0.784 (see the table in the Appendix A). Literature data are confirmed, both regarding previous Italian versions [42,43] and translations in other European and non-European languages [44].

#### 2.1.2. STAXI-2

The STAXI-2 [45] is a self-report scale that assesses anger as an emotional state of varying intensity. It consists of 57 items divided into six subscales (trait anger, state anger, internal/external anger expression, external/internal anger control), five subscales, and an anger expression index. Specifically, state anger (15 items) measures the intensity of feelings of anger and the extent to which a person is likely to express anger at a given time; whereas trait anger (10 items) measures the frequency with which feelings of anger are experienced to an excessive degree. In the scale adapted to the Italian population [46], Cronbach’s alpha is 0.84; in our sample, Cronbach’s alpha is 0.865.

### 2.2. Sample

The sample consists of 428 adult subjects.

In the demographic data, participants were asked to indicate their biological sex; there were no people who refused to answer the question or who entered other specifications.

The male sample consisted of 99 subjects; the age range was 24–64 years, with μ = 49.10 and SD = 9.25. In terms of education, 53.5% of the sample had completed high school and 41.4% had completed university. A total of 56.6% had children. In terms of occupation, the largest proportion (35.4%) were administrative and cultural officers, followed by 30.3% police officers and 20.2% professionals. Managers made up 4% of the male sample.

The female sample consisted of 339 respondents; the age range was 22–65 years, with μ = 47.22 and SD = 9.74. A total of 62.8% had children. In terms of education, 39.5% of the sample had completed secondary school and 56.6% had a university degree. With regard to their occupation, the majority (48.7%) worked in administration and cultural services, one fifth (20.1%) were kindergarten or nursery schoolteachers, 12.4% were social workers, with fewer people working as police officers (5.3%), as professional and technical workers (5.9%) or as labourers and drivers (3.5%). A total of 3.2% of women are in managerial positions.

### 2.3. Statistical Analysis

Data were described as frequencies, medians, arithmetic means, and standard deviations. Given that the scales are not interval scales and that the sample size was not always large, the use of non-parametric tests was preferred. The Mann–Whitney test and the Kruskal–Wallis test were used to assess the differences between the independent variables; the ANOVA test was used only on the whole sample to ensure greater robustness of the analysis; Spearman’s correlation coefficient was used to measure the relationship between the variables analysed. The level of significance used in the analyses was *p* = 0.05. All statistical calculations were carried out using IBM SPSS Statistics version 26. Covariance analysis was also performed, considering the various basic affective systems in relation to sex and age. The STATA programme was used for this analysis.

## 3. Results

Although there was a substantial numerical difference between the female and male sample, comparisons were possible due to the homoscedasticity of the samples. In fact, Levene’s test was not significant either when considering age (*p* = 0.271–0.241) or when assessing the groups with respect to ANGER (*p* = 0.875–0.890), FEAR (*p* = 0.061–0.056), SADNESS (*p* = 0.437–0.460), state anger (*p* = 0.334–0.219), and trait anger (*p* = 0.282–0.247).

A first starting point is to assess the differences between the affective systems in relation to sex. The distribution of scores according to biological sex differs for the following emotions: CARE [F (1-436) = 37.65, *p* = 0.000] SADNESS [F (1-436) = 16.95, *p* = 0.000], (Social) Dominance [F (1-436) = 5.79, *p* = 0.017], Social Anxiety [F (1-436) = 9.24, *p* = 0.003]. Although it is possible to apply ANOVA even when there is not always a normal distribution due to the sample size, we also report the results of the Mann–Whitney U-test comparison, which confirm the previous data, with the following statistical significance: CARE (*p* = 0.000), SADNESS (*p* = 0.000) Social Anxiety (*p* = 0.006), Dominance (*p* = 0.000). No significant differences were found for ANGER and FEAR. Table 2 and Table 3 show the means, standard deviations, and medians of the differentiated primary and secondary affective systems for women and men.

For anger, both state and trait, there were no statistically significant differences between the male and female samples.

Regarding the second objective, there are clear correlations (*p* = 0.01) between FEAR, SADNESS, and ANGER, in both the female and male samples, but with differences in the magnitude of the correlations, detailed as follows: the strongest correlation, in both women and men, is between FEAR and SADNESS, but this correlation has a greater magnitude in women, while the correlations with ANGER have a greater magnitude in men than in women. On the STAXI-2, the correlations are significant, with a greater magnitude for the trait anger (Table 4, Table 5 and Table 6).

To better understand how FEAR, ANGER and SADNESS may influence or be influenced by other affective systems, the female and male samples were divided into four groups according to the quartiles of the FEAR, ANGER and SADNESS scores. The different groups were evaluated in terms of the scores for the different affective systems and, in the case of statistically significant differences, the groups that differed were identified. Table 7, Table 8 and Table 9 show the significant systems and pairwise comparisons in order of significance (from *p* = 0.01 to *p* = 0.05). Table 10 and Table 11 show the mean, median, and standard deviation of affective systems compared to the different quartiles of SADNESS and FEAR.

The SEEKING distribution is not statistically different in the different quartiles of SADNESS, neither in the female nor in the male sample.

For both women and men, increasing levels of ANGER correspond to parallel increases in FEAR; also, SADNESS, Dominance, and Social Anxiety have the same consistent trend with ANGER. The CARE system, on the other hand, shows the opposite trend, decreasing as anger increases. ANGER, therefore, seems to involve the same affective systems and trends in the same direction, regardless of biological sex.

Increased levels of FEAR are matched by similar changes in the systems of ANGER, Social Dominance, SADNESS and Social Anxiety scores for both women and men. Some systems show significant differences only in the female groups. The highest scores in SEEKING and PLAY are expressed by the subjects who have the lowest scores in FEAR, i.e., who belong to the first quartile. (Other significant differences with respect to quartiles are detailed in Table 9). CARE decreases from group 1 to 2 and then increases; the highest mean in the CARE scale scores is expressed by the fourth quartile group.

Looking at the groups differentiated by SADNESS scores, there is a gradual increase in FEAR and ANGER in both males and females. It is interesting to note that only in the male groups is there a parallel increase in Dominance, but only up to group 3; a further increase in SADNESS (group 4) corresponds to a slight decrease in Dominance scores. Regarding SADNESS, it can also be observed that in the female sample there is a significant difference between the CARE and PLAY scores, which show a divergent trend between them; PLAY decreases and CARE progressively increases.

In short, when comparing the groups of the different quartiles with respect to the ANGER scores, there are variations in the same basic affective systems in both the female and male samples; comparing the quartiles of the SADNESS, in addition to differences in the FEAR, and ANGER systems, there are variations that affect only the male sample—Dominance—and variations that affect only the female sample—CARE, PLAY and Social Anxiety; finally, when comparing the groups of the different quartiles with respect to FEAR, in addition to differences in the ANGER, SADNESS, and Dominance systems, changes in the SEEKING, CARE and PLAY scores are evident in the female sample.

We point out, given the wide age range of our sample, that there is no interaction between sex and age in the analysis of covariance (see the Appendix A). Instead, age influences the female and male sample equally, leading to a decrease in FEAR scores (*p* < 0.001), ANGER (*p* = 0.003), PLAY *(p* = 0.01) and Dominance *(p* = 0.019).

For the state of anger, comparing samples with scores below or above the 50th percentile, the differences between the sexes are confirmed. Male subjects with higher scores in the state of anger also have higher scores in FEAR (*p* = 0.011), and in ANGER (*p* = 0.020). In the female sample, on the other hand, the affective systems that show significant differences are ANGER (*p* = 0.006), SEEKING (*p* = 0.047), PLAY (*p* = 0.025), SADNESS (*p* = 0.009), and Social Anxiety *(p* = 0.013).

With regard to the trait anger, FEAR was statistically significant (*p* = 0.000) in the female sample but not in the male sample (*p* = 0.060), while the other systems that show significant differences are the same for males and females, listed as follows: ANGER (females *p* = 0.000; males *p* = 0.000), SADNESS (females *p* = 0.000, males *p* = 0.008), Social Dominance (females *p* = 0.010, males *p* = 0.003), Social Anxiety (females *p* = 0.019, males *p* = 0.034) (Table 12, Table 13, Table 14 and Table 15).

## 4. Discussion

Regarding sex differences in basic affective systems, the data from our sample only partially support what has been found in the literature. In the female sample, scores for the SADNESS and CARE systems are higher than in the male sample, which is in line with what has been found in the literature, also considering studies on mammals, which demonstrate a genetic and hormonal basis (oxytocin and progesterone) for these differences [33,41,47,48,49,50,51,52]. On the other hand, comparing the samples, there are no differences in the ANGER and FEAR systems. Regarding the ANGER system, its predominance in male humans would be supported by both anatomical and hormonal perspectives [33,35,53], but a recent review [54] surprisingly found no gender difference in 13 out of 15 countries, which led the authors to question the possible discrepancy between behaviour and self-reporting, as well as the complexity of this emotion. Our sample is therefore consistent with these new data. Furthermore, the lack of significant differences between male and female samples is also confirmed in the STAXI-2 data, both in terms of state and trait. Regarding the FEAR system, in the same review, in most North American and European countries, women show higher scores than men; in our sample, however, there is no statistically significant difference. We have already mentioned that there is an important genetic and hormonal component in the expression of affective systems, but there is also a cultural modulation [53]. We can observe that the CARE and SADNESS systems are active in attachment relationships, which probably refer to situations in which there is still a substantial gender imbalance. In Italy, caring for children and the elderly is still characterised as a predominantly female task. It is therefore not unexpected to find higher scores in the female sample in our sample. ANGER and FEAR may perhaps be expressed in contexts in which sex differences are less marked; furthermore, the specific characteristics of the working environment must be considered. As a rule, employment in a municipality is associated with job security and non-stressful working conditions; on the other hand, the possibility of career advancement and the contestation/fight for job advancement are less strong. Furthermore, it should be added that those that are highly competitive and eager to make their way are more likely to choose larger private companies or more prestigious public institutions. It seems to us that the lack of differences between the female and male samples, with respect to FEAR and ANGER, can be interpreted partly as a sign of less cultural rigidity with respect to some gender differences, and partly as linked to the specific environmental and work situation.

As for the second objective, FEAR, SADNESS, ANGER, state, and trait anger seem to be not so distant from each other. This is in line with the affective neuroscientific literature, also considering Italian samples [41,42,43]. According to affective neuroscience, the affective systems FEAR, ANGER and SADNESS have different anatomical localizations (RAGE: from the medial amygdala to the bed nucleus of the stria terminalis (BNST), medial and periform, hypothalamus PAG; SADNESS: anteriorcingulate BNST POA dorsomedial thalamus PAG; FEAR: from the central and lateral amygdala to the medial hypothalamus and dorsal PAG) and different neuromodulators (RAGE: Substance P, acetylcholine, glutamate; SADNESS: opioids, oxytocin, prolactin, corticotrophin-releasing factor (CRF), glutamate; FEAR: glutamate, diazepam-binding inhibitor (DBI), corticotrophin-releasing factor (CRF), cholecystokin (CCK), alpha-MSH, neupeptide Y) [55]. As can be seen, glutamate is common in all three affective systems, but above all, all three relate to “negative affects” that, albeit in different ways, signal the organism to activate for its own protection. It is not surprising, therefore, that they are interrelated. Experience (learning) will then teach us which system should have priority in a given circumstance and which behaviour to display. For example, it is not always useful to attack, even when we feel ANGER; if we are in a disadvantageous position, attacking would cause FEAR, and it is therefore better to inhibit aggressive behaviour. The SADNESS system itself consists of an initial protest response, which we might consider anger, followed by a depressive phase of anxiety or fear of abandonment if reunion with the caregiver does not occur. We can also consider the case of paranoia, a particular type of fear that is not realistically well-founded. Paranoia could derive from anger, from a tendency to attack that clashes with fear; paranoia can be understood as a state of anxiety that arises from the conflict between anger and fear.

The peculiarity of our sample, in which no statistically significant differences between the female and male samples can be inferred with respect to ANGER, FEAR, and state and trait anger scores, makes a comparison with the activation of other basic affective systems particularly interesting.

As far as the state of anger is concerned, males and females respond differently, and what is most striking is the number of systems in the female sample in which significant differences are recorded. Whereas in males the only differences concern FEAR and ANGER (two close systems), in females ANGER, SADNESS, Social Anxiety, SEEKING and PLAY are involved. One question we can ask ourselves is whether the reason for the state of anger is the same for men and women: anger that arises in response to an obstacle that stands in the way of one’s goal might evoke a circumscribed response, namely ANGER and/or FEAR, but if the state of anger were motivated by relational contrasts, perhaps the response would be broader and more articulated. Perhaps this could be a hypothesis for interpreting such differences between men and women. Partial support may come from comparisons with trait anger, where high or low levels lead to variations in the same systems in men and women, with the sole exception of FEAR, which does not vary in the male sample, but only in the female sample. Even when comparing groups with increasing levels of ANGER, there is no difference between the male and female samples; in both, there is a consistent increase in the same affective systems (FEAR, SADNESS, Dominance) and a decrease in CARE, with no difference in biological sex.

Significant differences emerge when looking at FEAR levels.

It seems relevant to us to observe how, as FEAR increases, there are differences in the scores for ANGER, Dominance, SADNESS and Social Anxiety in both women and men, but only in the female groups are there significant differences in the SEEKING, CARE and PLAY systems.

It is important to note that, starting with the subjects in the second quartile with respect to FEAR scores, a significant reduction in scores related to the SEEKING system is noted. This is relevant because the SEEKING system is active by default and is transversal to all others. Any desire the subject wants to satisfy must, in fact, be sought in the external world. One can therefore understand the great significance of the hypoactivation of the SEEKING system. It should be mentioned here, to reinforce the observation, that the sample was not clinical. No less relevant in terms of quality of life is the progressive hypoactivation of the PLAY system, which in adults finds greater expression in the management of social roles, and in the pleasant connotation of various activities.

Another interesting finding, which concerns only the female sample, is related to the theme of care. Subjects whose FEAR and SADNESS scores are in the first quartile (i.e., low scores) have higher scores on the PLAY scale, while the mean highest scores in CARE are expressed, for both the FEAR and SADNESS scales, by the fourth quartile group. This could suggest that when the caregiver is in state of serene balance, the activation of CARE, which is expressed in particular towards their sons and daughters, is associated with a positive and joyful dimension; that is, we could think of those situations in which mutual pleasure is perceived in the parent–child interaction. On the contrary, when the caregiver is experiencing personal difficulties (the activation of SADNESS or FEAR, i.e., fourth quartile) in taking care of others (CARE), the interaction risks losing the characteristic of mutual pleasure, of PLAY, and this is an impoverishment of the relationship. We could therefore advance the hypothesis that when the caregiver is experiencing personal difficulties, e.g., is afraid (FEAR), or does not feel safe and needs protection (SADNESS), taking care of others (CARE) may be a way to respond to their own needs. There is probably also a share of vicarious gratification in this, an attempt to satisfy in others what we cannot satisfy directly in ourselves [56].

It is interesting to note that the hypoactivation of the SEEKING system is only significant with increasing FEAR scores, and not with increasing SADNESS scores. It would therefore seem that FEAR is more pervasive and generalised to different aspects of life, putting the subject in state of withdrawal or alertness; SADNESS, on the other hand, seems to leave open the possibility of seeking, as if the subject is not resigned to not seeing his need for protection and care satisfied, and continues to seek. This, it should be emphasised, can be hypothesised regarding our data, which refer to conditions of relative normality; it should be verified whether this hypothesis can also be applied in situations of greater impairment of the SADNESS system. Furthermore, it should be remembered that activation does not define the success or effectiveness of the search.

It seems to us that these data may have relevant implications for health workers, who have been increasingly exposed to risks of violence in recent years and may therefore find themselves living with fear in the workplace. Violence in the workplace has become increasingly relevant and transversal across countries [57,58,59,60]. In European countries, 4% of health personnel reported that they had been subjected to verbal or physical workplace violence and harassment from patients or visitors [61]. Although the vast majority of workplace violence and harassment is not physical, it can have significant psychological effects, such as symptoms of anxiety, depression, post-traumatic stress disorder, hypervigilance, suspicion, exhaustion, discomfort, job dissatisfaction, and burnout [62,63]. We observe how at least some of the symptoms listed can be consistently interpreted in the light of affective neuroscience theory. From this perspective, subjects with depressive disorders are characterised by an overactivation of the SADNESS system, and a hypoactivation of the SEEKING [64,65], and PLAY [66] systems. Moreover, since anxiety symptoms are strongly associated with depressive symptoms, there is generally a high activation of the FEAR system [67]. In patients diagnosed with Major Depressive Disorder, the involvement of the SEEKING, PLAY, FEAR and SADNESS systems is confirmed with the addition of the ANGER system, which is more activated [68].

By broadening the observations towards a more clinical context, future lines of research can be identified, particularly in the study of the factors involved in depression and the different impact they may have with respect to the sexes. It is known that various forms of clinical depression, and also the levels of depressive mood, have a higher incidence and prevalence in the female population compared to the male population, visible from puberty [69]. As far as is possible, limited to our non-clinical data, we can highlight a greater activation of SADNESS in the female sample, which could support a greater vulnerability for depressive aspects. In addition, there are variations in SEEKING and PLAY in relation to higher FEAR scores. We emphasise that, in our sample, there is no difference in FEAR scores in relation to gender, but what does change are the variations in SEEKING and PLAY. It seems to us, therefore, that FEAR, in addition to being evaluated for anxiety aspects, could be investigated as a somewhat co-etiological element and perhaps also as one of the possible foci in the treatment pathway. Perhaps different tools for managing fear and living and working environments that do not solicit this emotion could contribute to reducing depressive symptoms in the female population.

Our study has limitations due to the number and characteristics of the sample. In particular, the male sample is numerically smaller than the female sample. It would therefore be advisable to replicate the study on larger and more balanced samples. Another limit is noted in the characteristics of the workplace where the subjects of the sample work, albeit with different roles and tasks. In addition, not all employees with PC workstations responded and the dropout rate (around 50%) may have resulted in a more balanced sample. Finally, the instruments used, ANPS 3.1 and STAXI-2, are self-reports, and these have at least three types of limitations. The subject expresses his or her own assessment, which does not necessarily fully correspond to reality, in view of aspects that might escape his or her awareness. There may also be a modulation of responses according to social desirability. Finally, and this is an important aspect, the questionnaires presuppose a correct understanding of the contents of the items, which may not be the case. An method of reducing the risk of linguistic formulations that lend themselves to ambiguous interpretations with a consequent decrease in the validity of the questionnaires could have been the adoption of the natural language processing (NLP) method.

Of particular interest, however, are the differences in the activation of the affective systems SEEKING and PLAY in relation to FEAR in the female sample. The data require further research and insights, but they are very significant elements in the assessment of women’s well-being and gender equality. Certainly, gender inequalities have deep roots and causes, but a small contribution can also be made through the possibility of guaranteeing women safe environments. Increasing the sense of security in life and work contexts (decreasing FEAR) could increase teamwork (team play), provide a pleasant connotation of various activities (PLAY), an investment in new goals, and the search for new solutions (SEEKING); these are important elements both for well-being and as significant factors in promoting a working career. Field research taking these parameters into account may or may not confirm these hypotheses and possibly provide indications of good practice.

## 5. Conclusions

In summary, this research does not confirm a gender difference that would see males more inclined to respond with anger while women with fear. Furthermore, FEAR, SADNESS, ANGER, and anger in the trait and state assessments, are correlated and not opposed. The increase in ANGER is accompanied by the activation of identical affective systems in men and women, while there are differences between the sexes in the case of FEAR and SADNESS. The highest scores in PLAY were expressed by women who had the lowest scores in FEAR and SADNESS; the highest scores in SEEKING were expressed by women who had the lowest scores in FEAR. The fulfilment of life, considering the personal, working, and affective aspects, requires the activation of SEEKING and PLAY; we therefore highlight the need to monitor the conditions that, by activating FEAR and SADNESS, can lead to a hypoactivation of these affective systems. We believe that gender equity can also be achieved through the possibility of guaranteeing women the conditions in which their potential can be expressed, through the activation of SEEKING and PLAY.

## Figures and Tables

**Table 1 ijerph-21-01266-t001:** The salient features of the different affective systems.

Basic Affective Systems	Characteristics
CARE	We need to take care of others, especially the little ones, and especially our own offspring. It controls responses associated with nurturing behaviours and feelings and with the development of interpersonal relationships.
SEEKING	We feel the need to engage in searching for something, facing the problems that the world poses; all our biological appetites (including bodily needs such as hunger and thirst) can only be satisfied by the world. This is a foraging or searching instinct. It is perceived as interest, curiosity, and similar responses. It stimulates activities related to the exploration of the world, interest in reality, and seeking and anticipating positive experiences.
LUST	We feel the need to turn to sexual partners. This is perceived as sexual desire and arousal (sensuality).
PLAY	It is the medium through which one experiences the delimitation of one’s territory and its defence, through which social hierarchies are formed and where boundaries are formed and maintained within and outside the group. It controls responses related to social adaptation, the formation of social patterns, and prosocial attitudes.
PANIC/SADNESS	We feel the need to remain safe because of the presence of a caregiver (which is then internalised). Separation from attachment figures is initially perceived as agitation (protest), and later their loss is felt as despair. Expected state: my caregiver must be available and attentive to me.
FEAR	We feel the need to escape from (or paralyse ourselves during) threatening situations. Expected state: nothing should threaten my life and body.
ANGER/RAGE	We feel the need to attach and get rid of frustrating objects (things that stand between us and the satisfaction of our needs). Expected state: nothing should come between me and the satisfaction of my needs.

**Table 2 ijerph-21-01266-t002:** Means, standard deviations, and medians of primary affective systems in the female sample and male sample.

Sex		SEEKING	FEAR	CARE	ANGER	PLAY	SADNESS
Females	N	339	339	339	339	339	339
	Mean	45,83	37.58	45.51	29.47	38.32	37.45
	SD	6.658	9.443	7.354	8.926	8.177	7.594
	Median	46.00	37.00	46.00	30.00	37.00	37.00
Males	N	99	99	99	99	99	99
	Mean	45.10	36.39	40.42	29.51	39.27	33.75
	SD	7.534	10.505	8.692	8.899	8.936	8.777
	Median	45.00	35.00	40.00	29.00	41.00	34.00
Total	N	438	438	438	438	438	438
	Mean	45.67	37.31	44.36	29.48	38.54	36.61
	SD	6.864	9.693	7.957	8.910	8.353	8.018
	Median	46.00	37.00	45.00	29.00	38.00	36.00

**Table 3 ijerph-21-01266-t003:** Means, standard deviations, and medians of Social Dominance and Social Anxiety in the female sample and male sample.

Sex		Dominance	Social Anxiety
Females	N	339	339
	Mean	15.28	6.37
	Deviation std.	4.773	1.826
	Median	15.00	6.00
Males	N	99	99
	Mean	16.61	5.72
	Deviation std.	4.946	2.005
	Median	16.00	6.00
Total	N	438	438
	Mean	15.58	6.22
	Deviation std.	4.838	1.885
	Median	16.00	6.00

**Table 4 ijerph-21-01266-t004:** Values of Spearman’s correlation coefficient for ANGER, SADNESS, FEAR, state anger, and trait anger (*p* = 0.01).

	ANGER	SADNESS	FEAR	State Anger
SADNESS	0.306 **			
FEAR	0.396	0.629		
State Anger	0.236	0.164	0.199	
Trait Anger	0.570	0.324	0.334	0.303

** = *p* = 0.01.

**Table 5 ijerph-21-01266-t005:** Values of Spearman’s correlation coefficient for ANGER, SADNESS, FEAR, state anger, and trait anger (female sample) (*p* = 0.01).

	ANGER	SADNESS	FEAR	State Anger
SADNESS	0.293 **			
FEAR	0.360 **	0.652 **		
State Anger	0.207 **	0.168 **	0.146 **	
Trait Anger	0.570 **	0.295 **	0.316 **	0.300 **

** = *p* = 0.01.

**Table 6 ijerph-21-01266-t006:** Values of Spearman’s correlation coefficient for ANGER, SADNESS, FEAR, state anger, and trait anger (male sample). (*p* = 0.01).

	ANGER	SADNESS	FEAR	State Anger
SADNESS	0.405 **			
FEAR	0.525 **	0.595 **		
State Anger	0.340 **	0.116	0.346 **	
Trait Anger	0.576 **	0.387 **	0.378 **	0.290 **

** = *p* = 0.01.

**Table 7 ijerph-21-01266-t007:** Quartiles of ANGER and affective systems with statistically significant differences—pairwise comparisons.

ANGER	FEAR	CARE	DOMINANCE	SOCIAL ANXIETY	SADNESS
FEMALES	1–3 (*p <* 0.000) 1–4 (*p <* 0.000) 2–4 (*p <* 0.000) 2–3 (*p* = 0.012) 1–2 (*p* = 0.018)	3–1 (*p =* 0.008) 4–1 (*p =* 0.025)	1–4 (*p <* 0.000) 1–3 (*p <* 0.000) 2–4 (*p =* 0.002) 2–3 (*p* = 0.003)	1–4 (*p =* 0.004)	1–3 (*p <* 0.000) 1–4 (*p <* 0.000) 1–2 (*p* = 0.004) 2–4 (*p* = 0.011)
MALES	1–4 *(p <* 0.000) 2–4 (*p* = 0.001) 1–3 (*p* = 0.002)	4–1 (*p* = 0.008) 4–2 (*p* = 0.022)	1–4 (*p* < 0.000) 1–3 (*p* < 0.000) 2–4 (*p* = 0.005) 2–3 (*p* = 0.014)	1–4 (*p* < 0.000) 1–3 (*p* = 0.004) 1–2 (*p* = 0.045) 2–4 (*p* = 0.046)	2–4 (*p* = 0.004) 1–4 (*p* = 0.007) 2–3 (*p* = 0.007) 1–3 (*p* = 0.010)

**Table 8 ijerph-21-01266-t008:** Quartiles of SADNESS and affective systems with statistically significant differences—pairwise comparisons.

SADNESS	FEAR	ANGER	DOMIN ANCE	CARE	PLAY	SOCIAL ANXIETY
FEMALES	1–3 (*p* < 0.000) 1–4 (*p <* 0.000) 2–4 (*p* < 0.000) 3–4 (*p* < 0.000) 1–2 (*p* = 0.001) 2–3 (*p* = 0.001)	1–4 (*p* < 0.000) 1–2 (*p* = 0.006) 3–4 (*p* = 0.008) 1–3 (*p* = 0.011)		1–4 (*p* = 0.005) 2–4 (*p* = 0.012) 1–3 (*p* = 0.047)	4–1 (*p* = 0.009) 4–2 (*p* = 0.018)	1–4 (*p <* 0.000) 2–4 (*p* = 0.001) 1–3 (*p* = 0.002) 1–2 (*p* = 0.015) 3–4 (*p* = 0.017)
MALES	1–3 (*p <* 0.000) 1–4 (*p* < 0.000) 2–4 (*p <* 0.000) 1–2 (*p* = 0.007)	1–4 (*p <* 0.000) 1–3 (*p* = 0.002) 1–2 (*p* = 0.007)	1–3 (*p <* 0.000) 2–3 (*p* = 0.005) 1–4 (*p* = 0.006)			

**Table 9 ijerph-21-01266-t009:** Quartiles of FEAR and affective systems with statistically significant differences—pairwise comparisons.

FEAR	ANGER	DOMINANCE	SOCIAL ANXIETY	SADNESS	SEEKING	CARE	PLAY
FEMALES	1–3 (*p* <.000) 1–4 (*p* < 0.000) 1–2 (*p* < 0.000) 2–4 (*p* < 0.000) 3–4 (*p* = 0.036)	1–3 (*p* = 0.005) 1–2 (*p* = 0.031) 1–4 (*p* = 0.034)	1–3 (*p* < 0.000) 1–4 (*p* < 0.000) 2–4 (*p* < 0.000) 1–2 (*p* = 0.003) 3–4 (*p* = 0.046) 2–3 (*p* = 0.047)	1–2 (*p* < 0.000) 1–3 (*p* < 0.000) 1–4 (*p* < 0.000) 2–4 (*p* < 0.000) 3–4 (*p* < 0.000) 2–3 (*p* = 0.004)	4–1 (*p* = 0.001) 2–1 (*p* = 0.010) 3–1 (*p* = 0.011)	2–4 (*p* < 0.000) 2–1 (*p* = 0.005) 3–4 (*p* = 0.018)	4–1 (*p* < 0.000) 3–1 (*p* = 0.002) 4–2 (*p* = 0.006)
MALES	1–3 (*p* < 0.000) 1–4 (*p* < 0.000) 2–4 (*p* = 0.001) 1–2 (*p* = 0.040)	1–3 (*p* < 0.000) 1–4 (*p* = 0.003) 1–2 (*p* = 0.020)	1–4 (*p* = 0.001) 1–3 (*p* = 0.003) 2–4 (*p* = 0.007) 2–3 (*p* = 0.016)	1–3 (*p* < 0.000) 1–4 (*p* < 0.000) 2–4 (*p* = 0.003) 1–2 (*p* = 0.007) 2–4 (*p* = 0.026)			

**Table 10 ijerph-21-01266-t010:** Quartiles of SADNESS and mean, median, and standard deviation of affective systems (female sample).

SADNESS QUARTILES		SAD NESS	SEEK ING	FEAR	CARE	AN GER	PLAY	DOMI NANCE	SOCIAL ANXIETY
1	N	83	83	83	83	83	83	83	83
	Mean	27.93	47.48	30.28	43.75	26.04	39.42	14.78	5.65
	Median	29.00	47.00	30.00	45.00	24.00	39.00	15.00	6.00
	SD	3.741	6.430	8.216	8.018	9.707	7.227	4.862	1.641
2	N	97	97	97	97	97	97	97	97
	Mean	34.85	45.38	34.89	44.73	29.59	39.20	15.46	6.24
	Median	35.00	45.00	35.00	45.00	30.00	38.00	16.00	6.00
	SD	1.294	6.552	5.898	5.843	7.950	7.483	4.437	1.766
3	N	81	81	81	81	81	81	81	81
	Mean	40.44	45.40	38.85	46.52	29.41	38.68	15.12	6.47
	Median	40.00	45.00	39.00	46.00	29.00	38.00	15.00	7.00
	SD	1.710	5.987	6.616	7.623	7.648	7.543	4.675	1.817
4	N	78	78	78	78	78	78	78	78
	Mean	47.72	45.09	47.37	47.31	33.06	35.71	15.76	7.18
	Median	47.00	45.00	47.00	47.00	34.00	36.00	15.50	7.00
	SD.	3.259	7.473	8.075	7.586	9.167	9.981	5.195	1.786

**Table 11 ijerph-21-01266-t011:** Quartiles of FEAR and mean, median, and standard deviation of affective systems (female sample).

FEAR QUARTILES		FEAR	SEEKING	CARE	ANGER	PLAY	SADNESS	DOMI NANCE	SOCIAL ANXIETY
1	N	74	74	74	74	74	74	74	74
	Mean	25.42	48.14	46.30	24.73	41.38	30.76	14.53	5.31
	Median	27.00	48.00	48.00	23.00	41.00	31.00	15.00	5.00
	SD	4.233	6.528	7.561	8.740	8.309	6.031	4.863	1.735
2	N	102	102	102	102	102	102	102	102
	Mean	34.07	45.50	43.75	29.34	38.71	35.60	15.66	6.18
	Median	34.00	45.00	44.00	29.00	39.00	35.00	16.00	6.00
	SD	1.976	6.133	6.481	7.838	7.321	5.175	4.513	1.662
3	N	82	82	82	82	82	82	82	82
	Mean	40.29	45.62	45.09	29.98	37.93	38.82	15.43	6.67
	Mediana	40.00	45.00	45.00	31.00	37.00	38.50	15.00	7.00
	SD.	1.746	7.116	7.273	8.819	7.601	6.112	4.659	1.641
4	N	81	81	81	81	81	81	81	81
	Mean	50.36	44.36	47.43	33.47	35.46	44.52	15.36	7.26
	Median	49.00	45.00	48.00	34.00	36.00	45.00	15.00	7.00
	SD	5.192	6.508	7.829	8.566	8.719	6.366	5.122	1.773

**Table 12 ijerph-21-01266-t012:** Mean, median, and standard deviation of FEAR, ANGER, SADNESS, Dominance, and social anxiety in groups with high and low trait anger scores (female sample).

Trait Anger Groups		FEAR	ANGER	SADNESS	DOMINANCE	SOCIAL ANXIETY
1	N	189	189	189	189	189
	Mean	35.94	26.16	35.97	14.62	6.16
	Median	36.00	26.00	36.00	15.00	6.00
	SD	8.975	8.238	7.356	4.088	1.792
2	N	150	150	150	150	150
	Mean	39.64	33.65	39.32	16.11	6.62
	Median	38.00	34.00	40.00	16.00	7.00
	SD	9.641	7.975	7.501	5.416	1.842
Total	N	339	339	339	339	339
	Mean	37.58	29.47	37.45	15.28	6.37
	Median	37.00	30.00	37.00	15.00	6.00
	SD	9.443	8.926	7.594	4.773	1.826

**Table 13 ijerph-21-01266-t013:** Mean, median, and standard deviation of Social Anxiety, SEEKING, PLAY, SADNESS, and ANGER in groups with high and low state anger scores (female sample).

State Anger Groups		SOCIAL ANXIETY	SEEKING	PLAY	SADNESS	ANGER
1	N	253	253	253	253	253
	Mean	6.22	46.25	38.89	36.74	28.72
	Median	6.00	47.00	39.00	36.00	29.00
	SD.	1.805	6.583	7.986	7.328	8.861
2	N	85	85	85	85	85
	Mean	6.76	44.71	36.85	39.47	31.73
	Median	7.00	45.00	36.00	39.00	31.00
	SD.	1.804	6.740	8.448	8.041	8.843
Total	N	338	338	338	338	338
	Mean	6.36	45.86	38.37	37.43	29.48
	Median	6.00	46.00	37.50	37.00	30.00
	SD	1.818	6.646	8.140	7.594	8.940

**Table 14 ijerph-21-01266-t014:** Mean, median, and standard deviation of ANGER, SADNESS, Dominance, and Social Anxiety, in groups with high and low trait anger scores (male sample).

Trait Anger Groups		ANGER	SADNESS	DOMINANCE	SOCIAL ANXIETY
1	N	58	58	58	58
	Mean	26.43	31.62	15.43	5.36
	Median	26.50	31.00	15.00	5.00
	SD	7.377	8.756	4.772	1.907
2	N	41	41	41	41
	Mean	33.85	36.76	18.27	6.22
	Median	34.00	35.00	18.00	6.00
	SD	9.131	7.977	4.759	2.056
Total	N	99	99	99	99
	Mean	29.51	33.75	16.61	5.72
	Median	29.00	34.00	16.00	6.00
	SD	8.899	8.777	4.946	2.005

**Table 15 ijerph-21-01266-t015:** Mean, median, and standard deviation of ANGER, and FEAR in groups with high and low state anger scores (male sample).

Trait Anger Groups		ANGER	SADNESS	DOMINANCE	SOCIAL ANXIETY
1	N	58	58	58	58
	Mean	26.43	31.62	15.43	5.36
	Median	26.50	31.00	15.00	5.00
	SD	7.377	8.756	4.772	1.907
2	N	41	41	41	41
	Mean	33.85	36.76	18.27	6.22
	Median	34.00	35.00	18.00	6.00
	SD	9.131	7.977	4.759	2.056
Total	N	99	99	99	99
	Mean	29.51	33.75	16.61	5.72
	Median	29.00	34.00	16.00	6.00
	SD	8.899	8.777	4.946	2.005

## Data Availability

The data presented in this study are available upon reasonable request to the corresponding author, because they are part of an ongoing study with other researchers.

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
