# Peer review of "Basic Affective Systems and Sex Differences in the Relationship between Anger and Fear"

_ijerph, 2024, doi:10.3390/ijerph21101266_

Round 1

Reviewer 1 Report

Comments and Suggestions for Authors

This research investigated sex differences in affective systems, and correlations between ANGER, FEAR and PANIC, state and trait anger in both a female and male sample. I have some following concerns:

1.     The title is too long. The title should briefly describe the main question. 

2.     Since no technology of neuroscience was used in the present study, “Affective neuroscience” in the title and abstract is misleading. I suggest the authors clarify that this is a behavioral study, and revise the ambiguous wording throughout the manuscript.

3.     The number of female subjects (339) is far more than the number of male subjects (99). The analyses of sex difference should be based on balanced number of female and male subjects. I suggest to collect more data of male subjects.

4.     The age range was 24-64 for males and 22-65 for females. The age is a confounding factor as age difference in emotion is widely tested. I suggest to take age as a Covariate in the analysis.

Reviewer 2 Report

Comments and Suggestions for Authors

I read the study titled Affective neuroscience and sex differences in the relationship between anger and fear. If FEAR increases, SEEKING and PLAY decrease, in the female sample Following an extensive review of the report, the observations are listed below.

·       The research presented in the document is original and accurately reflects the researcher's findings.

·       Title has to be shortened because it is too long. The conclusion section did not provide statistical evidence to support the claim that an increase in FEAR has an adverse impact on a decrease in SEEKING and PLAY.

·       A review of the literature was included in Section II: Materials and Methods. Please request the author to submit the scientific procedures, algorithms, etc. that were employed in getting at the research findings.

·       Using the questionnaires, data was gathered from 428 individuals. Could the author possibly clarify the questions that are used to collect data?

·       The process of assigning class levels to the gathered data is necessary to explain. Is the data format identical to that which is described in Table 01, with a citation at [35]?

·       Are all of the data that have been gathered accurate? If not, has any planning been done to prepare the data for analysis?

·       The data gathered might not be apparent! For instance, the phrase "I saw the man on the hill with a telescope" can signify either of the following:

o   I saw the man. The man was on the hill. I was using a telescope.

o   I saw the man. I was on the hill. I was using a telescope.

o   I saw the man. The man was on the hill. The hill had a telescope.

o   I saw the man. I was on the hill. The hill had a telescope.

o   I saw the man. The man was on the hill. I saw him using a telescope

·       Currently, the most popular modern technology and method for text data analysis and text mining is natural language processing (NLP). It is advised that the author take this method. At the very least, it could be noted as a potential future study if the author finds it difficult to use NLP for research findings.

Round 2

Reviewer 1 Report

Comments and Suggestions for Authors

I still have one concern. I think it is a big issue.

As there are far more female participants than male participants, it is not appropriate to compare sex differences. I suggest to remove the results of male participants and tell a story only about female.

Author Response

Dear Reviewer, 
even if the groups are numerically different, the comparison is possible if the variability of the residuals is constant, i.e. if there is homoscedasticity. I have therefore integrated the statistical information into the text.
 I have also asked for advice from the statisticians of the University of Verona, who confirmed the correctness of what has been reported. 

I also pointed out this imbalance as a limitation of the work, hoping that it could be replicated with numerically homogeneous samples.
Best regards